# Degradation of Dental Methacrylate-Based Composites in Simulated Clinical Immersion Media

**DOI:** 10.3390/jfb13010025

**Published:** 2022-02-28

**Authors:** Nicoleta Ilie

**Affiliations:** Department of Conservative Dentistry and Periodontology, University Hospital, LMU, 80336 Munich, Germany; nilie@dent.med.uni-muenchen.de; Tel.: +49-89-44005-9412; Fax: +49-89-44005-9302

**Keywords:** resin-based composites, aging, three-point bending, instrumented indentation

## Abstract

The selection of restorative materials with regard to the longevity and durability of a restoration is of crucial importance for daily dental practice and requires that the degradation of the material in the oral environment can be assessed. The aim of this study was to investigate the extent to which the mechanical properties of four (Esthet X, Ceram X, Filtek Supreme XT, and Filtek Supreme XT flow) resin-based composites (RBCs) alter during storage in saliva substitutes (artificial saliva) for 24 h and 28 days and in the context of simulated, more aggressive clinical conditions, including cycles exposure to de- and remineralization, alcohol, or salivary enzymes. For this purpose, flexural strength and modulus were determined in a three-point bending test (*n* = 20) followed by Weibull analysis, while quasi-static behavior was evaluated by instrumented indentation techniques. Degradation occurred in all RBCs and all aging protocols and was quantifiable at both macroscopic and microscopic levels. The postulated stabilizing effect on degradation through the incorporation of urethane-based co-monomers into the organic matrix or a higher filler loading is refuted. Even though modern RBCs show high clinical survival rates, biodegradation remains an issue that needs to be addressed.

## 1. Introduction

Resin-based composites (RBCs) are currently the most commonly used dental restorative materials and have been proven over decades to efficiently replace damaged tooth structure [1,2]. Cumulative survival rates of 91.7% at 6 years, 81.6% at 12 years, and 71.4% at 29 years clearly demonstrate the longevity of RBC fillings [1]. Dominant among these attributes is the synthesis versatility of the individual compounds, allowing the fabrication of a wide variety of materials, since RBCs, similar to the tooth structure they replace, are composed of organic and inorganic components that are interconnected. This leads to tailored properties of bulk materials, but also to local inhomogeneity and graded properties, as observed in light-cured RBCs [3,4]. By fine-tuning the chemical composition of each component, their distribution, interaction, and morphology, modern RBCs have achieved significant improvements in mechanical performance [5], bioactivity [6], and aesthetic [7] while simultaneously addressing their deficiencies such as polymerization shrinkage [8].

The harsh conditions that RBC restorations are exposed to in the oral cavity still place high demands on their strength and durability [9], while hydrolytic degradation remains one of the more difficult challenges [10]. The potential degradation of a monomer system is directly related to the inherent chemical stability of the atomic groups that make up the monomer [11]. The degradation potential, therefore, lies in the nature of the materials, since dental RBCs are predominantly based on dimethacrylates such as bisphenol A glycol dimethacrylate (Bis-GMA), ethoxylated bisphenol A dimethacrylate (Bis-EMA), urethane dimethacrylate (UDMA), triethylene glycol dimethacrylate (TEGDMA), or complex derivatives of these monomers, thus containing a number of hydrolysable bonds such as esters, ethers, urethanes, and amides, which can be cleaved by hydrolysis [12]. It has been clearly demonstrated that degradation occurs in many RBCs despite varied vinyl acrylate compositions [13].

Aside from chemical degradation by hydrolysis, a clear disadvantage of RBCs placed in the oral environment is the fact that hydrolytic degradation is catalyzed by salivary enzymes, a process defined as biodegradation [10,12,14]. It is mainly the type of bonding within the polymer backbone that determines the rate of hydrolysis. Since the rate constants of enzymatic hydrolysis measured for diverse (di)methacrylates used in RBCs differ [12], their chemical stability depends on the nature of the monomers of which they are composed [12]. Cholesterol esterase (CE) and pseudocholinesterase (PCE)-like hydrolase activities of human saliva have been shown not only to act at levels that could degrade RBCs [15], but also to act synergistically to enhance biodegradation of RBCs containing TEGDMA and Bis-GMA, after previously postulating that CE had a greater catalytic effect on the degradation of Bis-GMA and PCE on TEGDMA [16]. Another worrying aspect in addition to biodegradation is that toxic and/or probiotic products can also be released due to the susceptibility of the ester groups to hydrolysis by salivary enzymes. These include methacrylic acid (MA), triethyleneglycol (TEG), and bishydroxypropoxyphenyl-propane (BisHPPP) [17].

In addition, it needs to be considered that hydrolysis reaction rates through catalysis are strongly influenced by the pH of the degradation medium. For the hydrolysis of esters, for instance, it has been shown that catalysis can change the reaction rates by several orders of magnitude, while the hydrolysis can be either acid or base catalyzed [18]. These findings make it difficult to understand the degradation of RBCs in the oral cavity, since dietary components, enzymes from the saliva, and microorganisms constantly change the pH value of the medium to which the restorations are exposed. In addition to the pH of the degradation medium, the copolymer chemical composition and the amount and rate of water uptake are relevant factors for the degradation of a polymer matrix [12]. 

Moreover, the hydrolytic and biodegradation in RBCs is not limited to the polymer matrix, since the degree of adhesion of the organic polymer to the inorganic filler is also susceptible to chemical attack [19]. The silane coupling agents used therefore are of an amphiphilic nature that bear two different types of functional groups. One is an organic functional group (methacrylate) that provides adhesion to the polymer matrix through polymerization, and the other is a silicon functional group (generally alkoxy groups) that can react with the active sites of the inorganic fillers. Both the ester bond of the methacrylate and the siloxane bond tend to degrade in time in a humid environment, accelerating RBCs degradation [20]. This statement was clearly demonstrated in RBCs of similar chemical composition, in which increased filler loading resulted in improved bio-stability over time after exposure to cholesterol esterase [19]. 

Since RBCs biodegradation changes the mechanical properties, which in turn are clinically responsible for the durability of a restoration [21,22], the aim of the study was to quantify the effect of exposing various RBCs to degradation media that simulate oral fluids/conditions, based on the observations described above. The tested null hypotheses were that the degradation behavior of analyzed RBCs quantified by macro- (flexural strength, flexural modulus, and reliability) and micro-mechanical properties (quasi-static parameters of instrumented indentation test: indentation hardness, Vickers hardness, indentation modulus, and creep) has a similar impact irrespective of: (a) incubation medium (artificial saliva, pH cycling, alcohol cycling; enzyme); or (b) filler amount. 

## 2. Materials and Methods

Four light-cured, nano-filled, nano- and micro-hybrid RBCs were selected (Table 1). The same shade (A3) was chosen for all RBCs, which is a common shade for dental restoratives. The light exposure time was similar for all materials (20 s) to assure direct comparison, and a modern blue-LED (Light Emitting Diode) LCU (Light Curing Unit) was used for polymerization (Elipar Freelight 2, 3M, Seefeld, Germany). The LCU was applied directly to the sample surface protected by polyacetate sheets. The LCU’s irradiance was 1241 mW/cm^2^, as measured by a spectrophotometer (MARC, Managing Accurate Resin Curing system; Bluelight Analytics Inc., Halifax, NS, Canada).

A 3-point bending test was employed to determine the flexural strength, FS, and flexural modulus, E. The quasi-static behavior of the analyzed RBCs was monitored by an instrumented indentation test (FISCHERSCOPE^®^ H100C Helmut Fischer, Sindelfingen, Germany). RBC specimens were analyzed after exposure to 5 different storage conditions. An initial immersion in artificial saliva (pH 6.9; 1000 mL: 1.2 g potassium chloride, 0.84 g sodium chloride, 0.26 g dipotassium phosphate, 0.14 g calcium chloride dehydrate) for 24 h at 37 °C was commune for all 400 specimens. A fifth of these served as controls while the other 4/5 were allocated to additional immersion media (Figure 1). These included a 28-day immersion in artificial saliva at 37 °C, a 28-days cyclic de- and demineralization procedure, an alcohol/saliva cyclic procedure for the same period, and a 28-days immersion in an enzymatic solution (lysozyme). 

### 2.1. Three-Point Bending Test

FS and E were determined in a 3-point bending test in conformity with NIST No. 4877, while using a span of 12 mm [23]. Specimen preparation occurred according to ISO 4049:2009 [24]. For this purpose, 400 specimens (*n* = 100 per RBC) were made by filling a white mold made of polyoxymethylene with an internal dimension of 2 mm × 2 mm × 18 mm with the RBC paste. Two glass plates with polyacetate sheets in between were used to compress the material. The specimens were overlapped and polymerized for 20 s (LED LCU Elipar Freelight 2) from top and bottom as specified in ISO 4049:2009 [24]. 

After demolding and grinding (silicon carbide abrasive paper, grit size p1200, LECO Corporation, Lakeview Ave. St. Joseph, MI, USA), all specimens were immersed for 24 h at 37 °C in artificial saliva. A number of 80 test specimens (*n* = 20 per RBC) were then removed and considered as reference groups (group: 24 h saliva) while the remaining samples were divided equally among 4 additional aging methods. Accordingly, 80 specimens (*n* = 20 per RBC) were stored for 28 days in artificial saliva at 37 °C (group: 28-days saliva), while the medium was changed daily, with the exception of the weekend, where the solution from the previous day was retained. Another group of 80 specimens (*n* = 20 per RBC) was subjected to a cyclic de- and remineralization process for 28 days (group: De-/remineralization). These specimens were immersed for 2 h in a demineralization solution (pH 4.0, adjusted with sodium hydroxide; 1000 mL: 1.22 g calcium chloride dehydrate, 1.14 g sodium dihydrogen phosphate-monohydrate, 3.0 g acetic acid 100%, and 1 L purified water) followed by 22 h immersion in a remineralization solution [25] (pH 6.85; adjusted with sodium hydroxide; 1000 mL: 0.275 g sodium dihydrogen phosphate-monohydrate, 0.294 g calcium chloride-dehydrate, 0.03 mg sodium fluoride, and 1 L purified water) at 37 °C. This de- and remineralization cycle was repeated every weekday, while during the weekend specimens were kept in the remineralization solution. Another group of 80 specimens (*n* = 20 per RBC) was subjected to a cyclic artificial aging of 28 days at 37 °C involving aging in an alcohol solution. The alcohol solution was a mixture of 50% distilled water and 50% absolute ethanol (99.9%; pH = 7.2). Specimens were immersed in the alcohol solution for 2 h, then in artificial saliva for 22 h (group: alcohol). Similar to above, the cycle was repeated daily except on weekends when samples were stored in artificial saliva. The last 80 specimens (*n* = 20 per RBC) were subjected to an enzymatic degradation process for 28 days at 37 °C in a solution of 0.22 g/L lysozyme (Carl Roth, Karlsruhe, Germany) in artificial saliva (group: lysozyme). The pH value of 20 g lysozyme in 1 L water is specified in the safety data sheets as 3–3.8; the molar mass of lysozyme is 14,000 g/mol. The hydronium ion concentration in the 20 g/L lysozyme solution (=10^−pH^; pH 3–3.8) is then 0.001–0.000158. The calculated pH of the solution used in the present study (0.22 g/L) is then 4.96–5.76. The solution was changed daily, except on weekends when the samples were stored in the previous day’s solution.

To assess the FS, specimens were loaded at a crosshead speed of 0.5 mm/min in a universal testing machine (Z 2.5, Zwick/Roell, Ulm, Germany), while recording the force as a function of the deflection of the beam. The slope of the linear part of the force-displacement diagram was then used to assess E.

### 2.2. Quasi-Static Instrumented Indentation Test (IIT)

100 fragments (*n* = 5) obtained from the 3-point bending test were wet-ground with silicon carbide abrasive paper with increasing grit size p1200, p2500, and p4000 (LECO Corporation, St. Joseph, MI, USA). This was followed by polishing for 2–3 min until the surface had a shiny appearance (automatic grinding machine EXAKT 400CS Micro Grinding System EXAKT Technologies Inc., Oklahoma, OK, USA). A total of 10 individual indentations were made on each specimen (50 indentations per group; 1000 indentations in total), with the indentation load and depth being recorded simultaneously during the load-unload-cycle according to ISO 14577 [26]. An automated micro-indenter (FISCHERSCOPE^®^ H100C, Helmut Fischer, Sindelfingen, Germany) equipped with a Vickers diamond tip was used for the test. The indentation was performed force controlled and at room temperature; the test load increased within 20 s and decreased within 20 s with constant speed in the range 0.4 mN to 500 mN, allowing calculation of the elastic and plastic deformation. A dwell time of 5 s was set for the maximum force of 500 N while the change in indentation depth during this time was recorded to serve as a measure for the material’s creep (Cr). 

Indentation creates an impression with the projected indenter contact area (A_c_) determined from the force–indentation depth curve considering the indenter correction based on the Oliver and Pharr model and described in ISO14577 [26]. To calibrate the indenter area function, we used sapphire and fused silica. 

The indentation hardness (H_IT_ = F_max_/A_c_) is related to the resistance opposed by the material to plastic deformation, and can be converted to the more popular Vickers hardness (HV = 0.0945 × H_IT_). The universal hardness (or Martens hardness; = F/As(h)) was calculated by dividing the test force by the surface area of the indentation under the applied test force (As) and characterized both plastic and elastic deformation of the material tested. To calculate the indentation modulus, the slope of the tangent of the indentation–depth curve at maximum force was employed [26].

### 2.3. Statistical Analyses

A Shapiro–Wilk procedure was employed to test the distribution of the variables. As all variables were normally distributed, a parametric approach could be used. Multifactor analysis of variance was then used to compare measured parameters (FS, E, Vickers and universal hardness, indentation modulus, and creep) among analyzed materials and storage conditions. A comparison occurred while using one- and multiple-way analysis of variance (ANOVA) and Tukey’s honestly significant difference (HSD) *post hoc* test with an alpha risk set at 5% as well as Pearson correlation analysis. The effect of the parameters material (RBC) and aging method as well as their interaction terms were quantified by means of a multivariate analysis (general linear model). Within this analysis, the practical significance of each term is described by the partial eta-squared (η_P_^2^) statistic adapted from the ratio of the variation attributed to the effect. In this context, larger values of η_P_^2^ indicate a larger variation accounted for by the model (SPSS Inc. Version 27.0, Chicago, IL, USA). 

FS data have been additionally described by a Weibull analysis. The Weibull model can be described as an empirical expression for the cumulative probability of failure (*P*) at applied stress (σ) [27]:(1)$$P_{f}(\sigma_{c})=1-\exp\left[-{\left(\frac{\sigma_{c}}{\sigma_{0}}\right)}^{m}\right]$$

In this equation, $\sigma_{c}$ represents the measured strength, m the Weibull modulus, and $\sigma_{0}$ the characteristic strength, which is defined as the uniform stress at which the probability of failure is 0.63. The outcome of the double logarithm of this equation is: $\ln\ln\frac{1}{1-P}=m\ln\sigma_{c}-m\ln\sigma_{0}$. Following ln ln(1/(1−*P*)) is plotted versus ln $\sigma_{c}$, resulting in a straight line. The upward gradient of the line allow calculation of the Weibull modulus, *m*, while the logarithm of the characteristic strength is the result of the intersection of the line with the x-axes [27].

## 3. Results

The parameters measured in the three-point bending test are summarized in Table 2 and Figure 2. A multifactorial analysis reveals a significant (*p* < 0.001) and high influence of the analyzed parameters RBC and aging method as well as their interaction product on the outcome of the three-point bending test. The effect of the parameter RBC was greater compared with the parameter aging method, with the studied material affecting the E more (partial eta-squared η_P_^2^ = 0.718) than the FS (η_P_^2^ = 0.400). In contrast, aging had a greater impact on FS (η_P_^2^ = 0.315) than on E (η_P_^2^ = 0.041). The binary interaction product was also significant (*p* < 0.001) while it was lower (η_P_^2^ = 0.148 for E and 0.090 for FS).

At the initial level of storage, 24 h post-irradiation and storage at 37 °C in artificial saliva, a one-way ANOVA evidenced three homogeneous subgroups for FS data in the following ascending sequence: (CX, EX, *p* = 0.212) < (EX, FSXT_F, *p* = 0.136) < FSXT, while the materials differ more clearly in terms of the E in ascending order: FSXT_F < CX< EX< FSXT (*p* < 0.001). This sequence is directly related to the amount of filler incorporated in the analyzed materials (Table 1).

The change in the measured properties after various storage methods can be depicted in Figure 2a–c. As found in the multivariate analysis, individual materials change more in terms of FS than E during storage. The only material that retained FS under different storage conditions was EX (*p* = 0.109). In CX, FS decreased in all aging methods relative to 24 h salivary storage, while there was no difference in the impact of the latter ones (*p* = 0.107). A much stronger differentiation was found in FSXT and in particular in FSXT_F. Alcohol/saliva cycling, De-/remineralization cycling, and the enzyme solution altered FS similarly (*p* = 0.07) and more strongly in FSXT than storage for 28 days in saliva, which in turn significantly decreased FS compared with the control group (24 h saliva). In the flowable RBC, FSXT_F, differentiation was more pronounced and occurred in descending order of the impact: saliva 24 h and 28 days (*p* = 0.9) < saliva 28 days and De-/remineralization (*p* = 0.775) < De-/remineralization and lysozyme (*p* = 0.212) < lysozyme and alcohol (*p* = 0.461). E follows this trend, but with a more flattened effect.

While the one-way ANOVA found no differences in FS between storage conditions for EX, the Weibull distribution (Table 3, Figure 3a–d) showed reduced reliability (lower Weibull modulus m) after De-/remineralization, alcohol, and lysozyme storage. This observation was also made for CX, with the difference that storage in artificial saliva for 28 days resulted in a decrease in reliability when compared with the control group. Differences in reliability during storage were comparatively small for FSXT, while for FSXT_F, only storage in alcohol and lysozyme led to a significant decrease in reliability.

The parameters measured in the instrumented indentation test are summarized in Figure 4a–e. Similar to the three-point bending test, the parameters RBC and aging method as well as their interaction product revealed a significant (*p* < 0.001) and high influence on the measured parameters, with RBC exerting a significantly stronger influence compared with the aging method.

The analyzed parameter RBC revealed a very high effect on both hardness parameters (universal and Vickers) but also on the indentation modulus (η_P_^2^ = 0.953, 0.945, and 0.938, respectively), followed by creep (η_P_^2^ = 0.597). In contrast, these effects were milder for the parameter aging method (η_P_^2^ = 0.479 for the universal hardness, 0.395 for the Vickers hardness, 0.485 for the indentation modulus, and 0.246 for the creep), evidencing, however, that the aging method influenced the indentation modulus stronger among all analyzed parameters. The binary interaction product was also significant (*p* < 0.001) while it was lower (η_P_^2^ = 0.108—universal hardness, 0.093—Vickers hardness, 0.126—indentation modulus, and 0.117—creep).

There was an excellent correlation between the analyzed parameters and the filler weight and volume, while the correlation for filler weight was slightly better. The highest correlation between filler amount and IIT parameters was observed for the indentation modulus (Pearson correlation parameter = 0.916 for the filler weight and 0.828 for the filler volume), followed by the universal hardness (0.903 and 0.804, respectively) and the Vickers hardness (0.863 and 0.763), while there was a highly significant inverse correlation with creep (−0.702 and −0.632). 

The highest correlation between the measured parameters was observed for universal and Vickers hardness (0.989), universal hardness and indentation modulus (0.982), and Vickers hardness and indentation modulus (0.942). Creep was inversely correlated with the above parameters, while the best correlation was identified for creep and indentation modulus (−0.789) and was only slightly lower for the hardness parameters (−0.758 and −0.739).

## 4. Discussion

RBCs are dental restorative materials that are used extensively worldwide [28] and have established themselves as the dominant restorative material, primarily because of their aesthetic properties [29]. After much debate about their longevity compared with amalgam fillings, it is now accepted that RBC fillings are at least comparable, if not significantly superior, in this regard [29]. However, their lifetime is limited by premature degradation, as individual components and the connection between them are vulnerable to immersion media and harsh environmental conditions, as summarized in the introduction.

To determine the degradation of the analyzed RBCs after immersion in simulated oral fluids, a three-point bending test was chosen, since FS is one of the few properties that has been shown to correlate with the clinical performance of RBC restorations [30]. The decision to use the change in mechanical properties as a measure of degradation was based on further clear clinical observations pointing to the failure of RBC fillings in recent decades, mainly due to fracture [21]. Along with the mechanical properties measured macroscopically, the study design envisages for measurement of mechanical behavior in the microscopic range. The advantage of the IIT measurements lies in the low indentation depth (5–6 µm for the tested materials), which makes the result independent of defects that would initiate premature failure in the macroscopic range. The performed statistical multifactorial analysis of variance underlined this observation, as measured IIT parameters showed the degrading influence of the storage media on the RBCs to be more sensitive (higher partial eta-squared values) compared with the mechanical tests performed at macroscopic scale.

The immersion media chosen to simulate aging were strongly focused on clinical relevance and storage was always at 37 °C. Storage in artificial saliva for both 24 h and 28 days is often used in dental material research [31] to simulate the initial conditions (24 h post-polymerization) and the effects of aging. Although this represents a simplified version of storage in natural saliva, it was necessary in this study design due to the large number of test specimens and the frequent change of immersion media. Artificial saliva was preferred to distilled water, due to its more pronounced degrading effect on mechanical properties [32]. In the present study, artificial saliva was used either as a solvent for the enzyme, as a neutral medium during aggressive alcohol cycles, or as a medium for the 24 h and 28-day immersion control groups.

Subjecting materials to pH cycling by de- and remineralization solution is a method commonly used in tooth De-/remineralization protocols to induce artificial caries [25]. This aspect, as well as the fact that the hydrolysis reaction rates of esters present in all monomers of the analyzed RBCs are strongly influenced by the pH of the degradation medium [18], led to the selection of this storage protocol. As for storage in alcohol, this procedure not only showed a clear correlation with the clinical behavior of RBC restorations in early studies [30], but is a fluid that frequently comes into contact with the restorative material, either in food and drink, or in commonly used mouthwash solutions. Besides, the enzyme-catalyzed degradation was simulated by using lysozyme, since its influence on the mechanical properties in the context of degradation processes has not been investigated so far. Lysozyme is present in relatively high concentrations in saliva and is the main component of the acquired dental pellicle, with a role in regulating microbial colonization and protection against acid attack [33]. It is a single-chain polypeptide of 129 amino acids cross-linked with four disulfide bridges and hydrolyzes beta (1–4) linkages between N-acetylmuraminic acid and N-acetyl-D-glucosamine residues in peptidoglycan and between N-acetyl-D-glucosamine residues in chitodextrin. The activity of lysozyme is related to the pH value while it is active over a wide pH range (6.0–9.0) [34].

It should be noted that conspicuous discolorations or changes in translucency, which are considered a relevant degradation indicator after storage in different media [35], could not be detected in any of the samples.

The selection of the RBCs for the present study was aimed at special features in the structure of the materials. Therefore, with FSXT, a material was selected that falls within the smaller but commonly used category of RBCs, which are defined as nano rather than nano-hybrid RBCs. This designation derives from the fact that the fillers are either nanofillers or agglomerates of nano particles and therefore the material do not contain any compact glass fillers, as is usual in nano- and micro-hybrids. Nevertheless, the size of the nano-agglomerate fillers is in the microscale range and is therefore comparable with the compact glass fillers observed in the nano-hybrid RBCs [4]. FSXT was the analyzed RBC with the highest mechanical properties, which can be attributed to two different aspects. The significantly higher proportion of inorganic fillers made the far greater contribution to this and had a particular effect on the modulus of elasticity, which was significantly higher than that of the other materials analyzed, both in the macro and in the micro range. The second aspect is the UDMA content of the polymer matrix, a co-monomer known not only to improve mechanical properties but also to increase resistance to degradation [11,36]. FSXT and the ormocer CX were the only materials analyzed with a UDMA content. While the initial mechanical properties (24 h storage in artificial saliva) were high in FSXT, significant degradation occurred when the material was exposed to longer and more aggressive immersion media. This degradation was less evident in the measured IIT parameters, showing a significant decrease in the magnitude of the other materials analyzed but was clearly reflected in the FS, which showed a steeper drop than in other materials. While storage in saliva and in the De-/remineralization solution also showed a trend towards reduced FS, alcohol cycling and storage in lysozyme solution reduced FS significantly. The degradation observed in FSXT must be related to the larger filler-matrix interface that emerged in this material, since not only was the amount of filler higher than the other materials but also the filler dimension (nanofillers and micro-clusters) was smaller [4]. This interface is in particular prone to hydrolytic degradation [19]. In the same note, it has been shown that the osmotic pressure built up at the matrix–filler interface due to hydrolytic degradation of the filler results in induced cracking at this interface, which is an additional potential degradation mechanism from a mechanical point of view [37]. Such defects may be responsible for initiating fracture in the three-point bending test specimens at an earlier stage since during loading they can grow to a critical size. Another argument for faster degradation can be seen in that a large filler–matrix interface can also be a reason for higher water uptake as measured in nano-filled RBCs compared with nano-hybrid RBCs [38].

In order to quantify the influence of the amount of filler on the degradation behavior, the flowable version of FSXT was also selected, since it was postulated that with a similar chemical composition, differences in bio-stability could be specifically linked to the relative resin/filler distribution [19]. This cannot be confirmed in the present study, as both FSXT and FSXT_F degrades significantly with aging. The explanation of the gradation in FSXT_F can focus on the organic matrix since the amount of matrix is not only higher than in FSXT [19], but also contains no UDMA. A low content of inorganic fillers is unmistakably reflected in the modulus of elasticity, similar to that described above, in both measurement methods, i.e., in the macro and microscopic range. Both E and indentation modulus were the lowest among the materials analyzed. The relatively high FS in flowable materials is an often observed phenomenon [5], and is initially deceptive. This is due to the very high flexibility of such materials, which emphasizes the imperative to determine not only the FS but also the E in a three-point bending test for a clear assessment of the mechanical behavior.

Next, a nano-hybrid RBC (CX) was selected, which has the peculiarity of containing ormocer (organically modified ceramic) compounds. Ormocers are based on UDMA modifications [39], and are characterized by a reduced shrinkage due to the large matrix monomers which are less crosslinked [8]. They also contain polysiloxanes which form inorganic Si-O-Si networks. The polysiloxanes are then crosslinked with polyfunctional urethane and thioether (meth)acrylate [40]. Increased stability against degradation was expected due to the UDMA content [11,36]. This assumption was not confirmed, since similar to FSXT, a strong decrease in FS with aging was observed. The IIT parameters were also reduced with immersion duration, while the effect of the different aging treatments had no differentiated influence. The enlarged filler–matrix interface, which can also be seen as a result of the reduced filler size in this material, since it is a nano-hybrid, could be held responsible for this behavior.

The rationale for including EX as the fourth RBC in this study arose after a literature review which found that EX was stable compared with other RBCs after storage in food-simulating media and ethanol solution [41]. EX is described as a micro-hybrid RBC, and the absence of the nanofillers in this composition is what sets it apart compared with the other three materials analyzed. The higher stability of EX is indeed confirmed in the present study, at least with regard to the FS data. It was the only material analyzed that was able to maintain FS through all aging conditions. It should be noted that a deterioration was also quantified here, but was expressed only in a reduced reliability (Weibull parameter, m), which once again underlines the importance of such statistical evaluation for FS data [42]. The scatter of the measured FS data was well modeled in all materials by Weibull statistics, which was confirmed by the very high R^2^ (>0.85) values summarized in Figure 4a–e and Table 3. Of note, the decrease in reliability was less in EX than with other materials and was not observed with the milder 28-day saliva storage. In contrast, the very sensitive IIT parameters could register a significant decrease in measured properties between 24 h and 28-days of saliva storage, indicating only superficial softening.

Therefore, the tested null hypotheses that the degradation behavior of analyzed RBCs quantified by macro- and micro-mechanical properties has a similar impact irrespective of (a) incubation medium (artificial saliva, pH cycling, alcohol cycling, and enzyme) or (b) filler amount, are rejected.

## 5. Conclusions

The effect of aging after 28 days of salivary storage and different aging protocols was measurable for all materials but had different impacts. The initial mechanical properties, in particular the modulus of elasticity and the indentation modulus, were directly related to the amount of filler, while the degradation has to be considered as a multifactorial event. Contrary to what was reported in the literature, urethane content did not improve degradation stability, nor did higher filler loading. Overall, alcohol appears to be the most degrading storage medium, followed by enzyme solution (lysozyme), De-/remineralization solution, and finally 28-day artificial saliva.

## Figures and Tables

**Figure 1 jfb-13-00025-f001:**
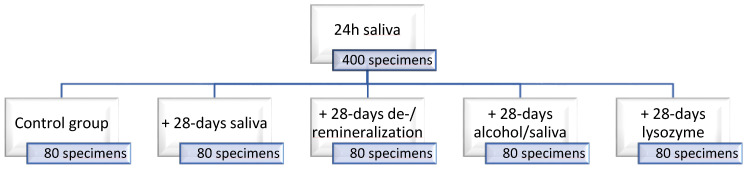
Schematic representation of the storage conditions and study design. With a number of 5 storage conditions and 4 materials, the total analyzed group number was 20.

**Figure 2 jfb-13-00025-f002:**
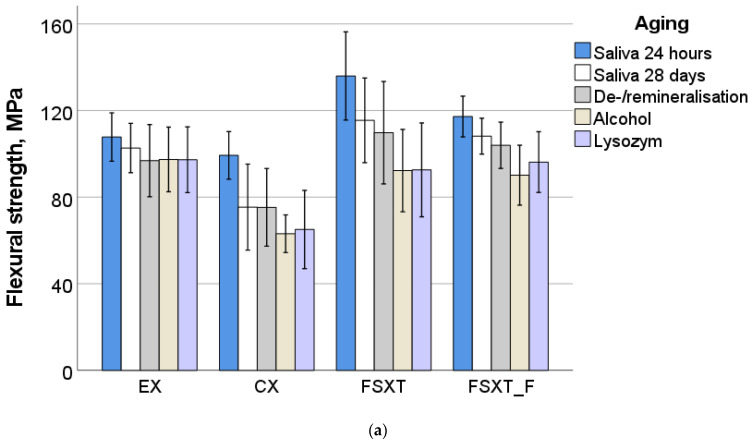
Three-point bending test: variation of (**a**) flexural strength and (**b**) flexural modulus with the aging method and analyzed material; (**c**) effect strength (η_P_^2^) of the parameters RBC and aging method as well as their interaction term on FS and E (*p* < 0.001).

**Figure 3 jfb-13-00025-f003:**
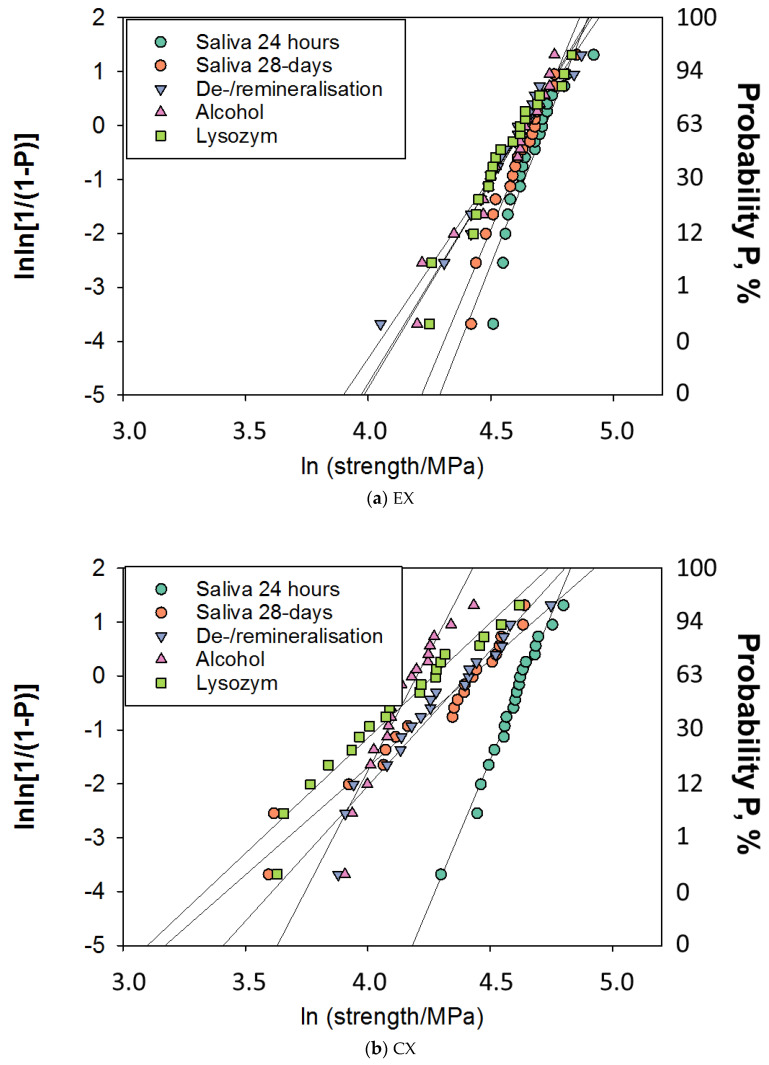
Weibull plot representing the empirical cumulative distribution function of strength data. Linear regression was used to numerically assess goodness of fit and estimate the parameters of the Weibull distribution, as summarized below in Table 3. (**a**) EX; (**b**) CX; (**c**) FSXT; and (**d**) FSXT_F.

**Figure 4 jfb-13-00025-f004:**
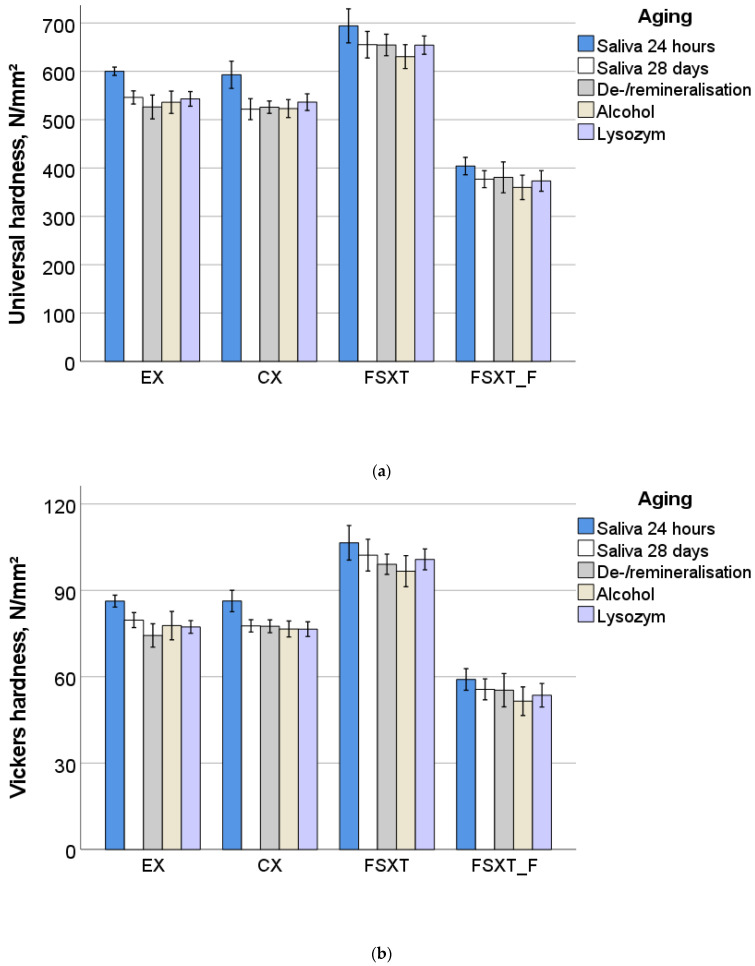
Quasi-static parameters measured in an instrumented indentation test (IIT) as a function of material and aging conditions: (**a**) universal hardness; (**b**) Vickers hardness; (**c**) indentation modulus, (**d**) creep; and (**e**) effect strength (η_P_^2^) of the parameters RBC and aging method as well as their interaction term on measured properties (*p* < 0.001).

**Table 1 jfb-13-00025-t001:** Analyzed RBCs: abbreviation (code), brand (LOT), manufacturer, and composition, as indicated by the manufacturer. The shade A3 was chosen for all RBCs.

Code	Material/LOT	Manufacturer	Monomer	Filler	
				Composition	wt/vol%
EX	Esthet X 0808001221	Dentsply De Trey	Bis-GMA, Bis-EMA, TEGDMA	SiO_2_, BaO–F–Al_2_O_3_–B_2_O_3_–SiO_2_	77/60
CX	Ceram X 0806003110	Dentsply De Trey	Bis-EMA, DM	Ormocer, BaO–Al_2_O_3_–B_2_O_3_–SiO_2_	76/57
FSXT	Filtek Supreme XT 20080905	3M	Bis-GMA, Bis-EMA, UDMA, TEGDMA	SiO_2_, ZrO_2_	78.5/59.5
FSXT_F	Filtek Supreme XT flow, 20080815	3M	Bis-GMA, Bis-EMA, TEGDMA	SiO_2_, ZrO_2_	65/55

Abbreviations: DM = diemethacrylate; SiO_2_ = silicon oxide (silica); ZrO_2_ = zirconium oxide; BaO–F–Al_2_O_3_–B_2_O_3_–SiO_2_ = barium–fluoro–alumino–boro–silicate glass; and ormocer = organically modified ceramic.

**Table 2 jfb-13-00025-t002:** Three-point bending test results for the reference storage (24 h, artificial saliva): mean and standard deviation for the flexural strength, FS and flexural modulus, E; data are arranged in ascending order of FS. Superscript letters designate groups without significant differences for FS and E data; Tukey’s HSD (honestly significant difference) *post hoc* test (α = 0.05).

RBC	FS, MPa		E, GPa	
	Mean	SD	Mean	SD
CX	99.3 ^a^	11.0	6.2 ^b^	0.5
EX	107.8 ^ab^	11.2	7.1 ^c^	0.7
FSXT_F	117.2 ^b^	9.4	5.3 ^a^	0.6
FSXT	136.0 ^c^	20.4	8.4 ^d^	0.9

**Table 3 jfb-13-00025-t003:** Parameters of the Weibull statistic as a function of RBCs and immersion conditions (m, Weibull parameter, SE = standard error; and R^2^ = coefficient of determination). The 95% confidence interval is defined as 1.96 × SE.

RBC	Storage	Weibull Parameters		
		R^2^	m	SE
EX	Saliva 24 h	0.88	11.56	1.00
	Saliva 28 days	0.95	10.84	0.59
	De-/remineralization	0.96	6.71	0.30
	Alcohol	0.96	7.44	0.36
	Lysozyme	0.95	7.61	0.41
CX	Saliva 24 h	0.99	10.83	0.38
	Saliva 28 days	1.00	3.99	0.20
	De-/remineralization	0.94	5.01	0.30
	Alcohol	0.90	8.76	0.69
	Lysozyme	0.95	4.27	0.22
FSXT	Saliva 24 h	0.98	7.69	0.27
	Saliva 28 days	0.98	6.88	0.26
	De-/remineralization	0.98	5.13	0.18
	Alcohol	0.95	5.09	0.28
	Lysozyme	0.98	4.89	0.18
FSXT_F	Saliva 24 h	0.91	13.69	1.03
	Saliva 28 days	0.98	15.90	0.56
	De-/remineralization	0.96	11.59	0.57
	Alcohol	0.97	7.53	0.30
	Lysozyme	0.85	6.55	0.64

## Data Availability

The datasets generated during and/or analyzed during the current study are available from the corresponding author on reasonable request.

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
