# Peer review of "Degradation of Dental Methacrylate-Based Composites in Simulated Clinical Immersion Media"

_jfb, 2022, doi:10.3390/jfb13010025_

Round 1

Reviewer 1 Report

Dear author,

thank you for the opportunity to review this all in all interesting and well-written paper. 

Subsequently, there are some points that need to be adapted, before it can be published:

Abstract: 

L.13:Please name the groups and the specific method used to assess degradation in the abstract.

Introduction:

L. 61: typo/grammatical error: should be "it needs to be considered" without comma before "that". 

L. 70-93: This two paragraphs should be moved in parts to the discussion part to short the introduction. 

L. 85-92: This part, from my point of view, belongs to the M6M section and should not appear in the introdcution. 

L. 106: Add "light" before exposure and add some information about the working distance during polymerization. 

L. 121: How was the pH-Value measured and adjusted?

L. 124: As the pH-value is a central point for biodegradation as described in your introduction, you should add information about the pH-value for every immersion medium.

L. 136: Maybe a figure would be helpful for readers not familiar with the method. Have you polymerized the whole cylinder of 18mm in one step? Or have you placed it in increments? 

L. 151: Can you provide an SOP for the de- and remineralizing procedures or has it been described in the literature before? To the reviewer, it seems quite uncommon to use a remineralization substance with pH below 7. 

L. 161: Please use a uniform spelling for lysozyme (or lysosym) during the whole manuscript and all figures. You should provide an information about the manufacturer or isolation procedure of the lysozyme. 

L. 191.: "a parameters that was indicated in the present study." Please correct the typo. 

The "statistical analyses" part is described exceptionally well. 

Results:

Tab. 2: Please rephrase the description of the meaning of superscript letters in the legend, as it is quite unclear.

Discussion:

In general, the discussion is very well written but extensive and should be shortened a bit. 

L. 335: You should discuss the possible effects of combining low pH-values and enzymatic activity as present in the oral cavity. 

What did you aim for to simulate using aggressive alcohol cycles?

In my opinion, the sentence "Despite this

degradation, in terms of FS data, FSXT was still comparable to the next best material in

terms of initial mechanical properties, which was EX." adds no value to the discussion and should be removed. 

L. 335: You should discuss the possible effects of combining low pH-values and enzymatic activity as present in the oral cavity. 

Author Response

All comments to the corresponding author have been addressed independently below. The authors’ rebuttal is always in BLUE and where changes have been added to the revised manuscript in light of the reviewer comments these are presented in RED.

The author would firstly like to thank the reviewers’ for taking the time to read and critically appraise the manuscript and secondly to thank the reviewers’ for their positive constructive comments in improving the work.

Comments and Suggestions for Authors

Reviewer comments:

Dear author,

thank you for the opportunity to review this all in all interesting and well-written paper. 

Subsequently, there are some points that need to be adapted, before it can be published:

 Author’s response:  Thank you for your comments and appreciation.

Abstract: 

L.13:Please name the groups and the specific method used to assess degradation in the abstract.

Author’s response:  thank you for this pertinent observation. I added the missing information, as suggested.

Introduction:

  1. 61: typo/grammatical error: should be "it needs to be considered" without comma before "that". 

Author’s response:  Thanks for pointing this out - I apologize for the error.

  1. 70-93: This two paragraphs should be moved in parts to the discussion part to short the introduction. 

The paragraphs mentioned was meant to explain that the degradation is not limited to the organic matrix but also to the silane, which is an important point reported in the literature.  The following paragraph was meant to prepare for the null hypothesis. The first paragraph has been reduced as suggested and the next deleted.   

  1. 85-92: This part, from my point of view, belongs to the M6M section and should not appear in the introdcution. 

Author’s response: as mentioned above, the intention was to help understanding the null hypothesis. I have deleted the paragraph, as suggested.

  1. 106: Add "light" before exposure and add some information about the working distance during polymerization. 

Author’s response: “light” was added, thanks or pointing this out

  1. 121: How was the pH-Value measured and adjusted?

Author’s response: the missing information on adjusting the pH value was added.  The solutions (de/remineralization, lysozyme, artificial saliva) were purchased and the pH was measured and adjusted at the factory/pharmacy.

  1. 124: As the pH-value is a central point for biodegradation as described in your introduction, you should add information about the pH-value for every immersion medium.

Author’s response: Thank you for pointing this out. I checked all pH-value. The pH-value of alcohol solution was not indicated, as alcohol is a neutral compound, so it's not acidic nor basic as it doesn't contain hydrogen ion nor hydroxyl ion. For the lysozyme solution the pH at the used concentration was calculated and indicated in the manuscript; for all other solutions the pH is indicated.

  1. 136: Maybe a figure would be helpful for readers not familiar with the method. Have you polymerized the whole cylinder of 18mm in one step? Or have you placed it in increments? 

Author’s response: Please note that the sample geometry is not a cylinder, but a slab of 2 mm x 2mm x 18 mm. The used curing method is a very commune and well known one in dentistry, as it is indicated in the mentioned ISO standard.

  1. 151: Can you provide an SOP for the de- and remineralizing procedures or has it been described in the literature before? To the reviewer, it seems quite uncommon to use a remineralization substance with pH below 7. 

Author’s response: yes, the used method for de and remineralisation are standardized methods used extensively in dental research, described in the literature and unanimously accepted. One of the most prominent authors are ten Cate J.M, with works started in the 80th on the topic. I added a reference therefor for more clarity. Please note that the pH of the remineralisation solution was almost 7; 6.85 to be exact.

  1. 161: Please use a uniform spelling for lysozyme (or lysosym) during the whole manuscript and all figures. You should provide an information about the manufacturer or isolation procedure of the lysozyme. 

Author’s response: I apologize for this inconsistency and revised accordingly; the mistake was predominantly in the figs. , which were all edited accordingly.

  1. 191.: "a parameters that was indicated in the present study." Please correct the typo. 

Author’s response: thank you for this observation

The "statistical analyses" part is described exceptionally well. 

Author’s response: thank you.

Results:

Tab. 2: Please rephrase the description of the meaning of superscript letters in the legend, as it is quite unclear.

Author’s response: I simplified the description for more clarity, please consider modification in the manuscript.

Discussion:

In general, the discussion is very well written but extensive and should be shortened a bit. 

Author’s response: I tried to reduce the words as good as possible, without reducing the content, as it is important to discuss the methods, materials and the results.

  1. 335: You should discuss the possible effects of combining low pH-values and enzymatic activity as present in the oral cavity. 

Author’s response: It is indeed a very challenging task to mimic the conditions in the oral cavity due to the extremely high number of factors involved. In order to understand the effects, parameters are first examined individually.  Too complex combination makes it difficult to understand the individual effects and leads to speculative assumptions.  Therefore, the effects were examined individually in the present study and everything was designed within the framework of feasibility.

What did you aim for to simulate using aggressive alcohol cycles?

Author’s response: As expressed in the discussion, a restorative material came into contact with alcohol either in food and drink or in mouthwash solutions, therefore I simulate these conditions in the present study.  Besides this clinical aspect, it has been clearly shown in the literature that aging RBC samples in alcohol in in-vitro tests correlates best with the clinical behavior of the restorations. The according literature is mentioned.

In my opinion, the sentence "Despite this degradation, in terms of FS data, FSXT was still comparable to the next best material in terms of initial mechanical properties, which was EX." adds no value to the discussion and should be removed. 

 Author’s response: I deleted the sentence - it was meant to point out that degradation occurred in FSXT, but the measured properties were not inferior to EX, despite degradation.

  1. 335: You should discuss the possible effects of combining low pH-values and enzymatic activity as present in the oral cavity. 

Author’s response: please consider answer above to the same suggestion.

Reviewer 2 Report

Authors compared the mechanical properties of four resin based composites suffered from aging condition in this study. I guess that the methods and results are appropriated. However, the following part should be improved;

  1. in figure4, the graph number is lost. The number should be described on each graph in figure 4.
  2. The value of each mechanical properties are directly shown in figure, but authors mention about larger values of partial eta-squared(ɳP2) in results part. However, the ɳP2 of each samples are not shown as table or figure. The value shown in figure and ɳP2 described in results are not matched, therefore, it is not easy to read the results. In term of ɳP2, it is better that the values are shown as a table or figure.

Author Response

All comments to the corresponding author have been addressed independently below. The authors’ rebuttal is always in BLUE and where changes have been added to the revised manuscript in light of the reviewer comments these are presented in RED.

The author would firstly like to thank the reviewers’ for taking the time to read and critically appraise the manuscript and secondly to thank the reviewers’ for their positive constructive comments in improving the work.

Comments and Suggestions for Authors

Authors compared the mechanical properties of four resin based composites suffered from aging condition in this study. I guess that the methods and results are appropriated. However, the following part should be improved;

  1. in figure4, the graph number is lost. The number should be described on each graph in figure 4.

Author’s response:  Thank you for pointing this out. I remediated the mistake – the figures are now each marked (a to d), as specified in the caption.

  1. The value of each mechanical properties are directly shown in figure, but authors mention about larger values of partial eta-squared(ɳP2) in results part. However, the ɳP2 of each samples are not shown as table or figure. The value shown in figure and ɳP2 described in results are not matched, therefore, it is not easy to read the results. In term of ɳP2, it is better that the values are shown as a table or figure.

Author’s response:  As described in the caption statistics, the partial eta-squared statistic reports the practical significance of each term, based on the ratio of the variation attributed to the effect. Whether this data match the figures or not is not possible to identify visually. It is a result of a statistical calculation to account for the effect strength and is not attributed to a sample. Based on your suggestion, I have added a table to figure 2 and one to the figure 4 (e), summarizing these parameters for better visibility. Thanks for this observation.

Reviewer 3 Report

The manuscript entitled “Degradation of dental methacrylate-based composites in simulated clinical immersion media” has been reviewed. The results are interesting and helpful. The manuscript needs to be well revised before acceptance. Detailed comments are as follows:

  1. There are some many typo errors in the manuscript. Units should be separated from the numerical value by a space. Please spell-check your manuscript.
  2. The abbreviation of resin-based composites should be RBCs.
  3. Some abbreviations, such as FS and E, should not be name twice.
  4. Pay attention to unnecessary capitalization of first letters of phrases, such as Urethane dimethacrylate and Triethylene glycol dimethacrylate.
  5. The structuring of 2. Materials and Methods is not adequate and must be improved. The first two sentences are methods (Line 101-104). The paragraph in Line 164-168 should be after the first paragraph of Three-point bending test.
  6. Please unify hours and h in the manuscript.
  7. The footnotes of Table 1 should be removed since these abbreviations are named before and the chemical structures are known.
  8. In Figure 3, the unit of x-axis is missing.
  9. In Fig. 4, Hardness should be hardness.

Author Response

All comments to the corresponding author have been addressed independently below. The authors’ rebuttal is always in BLUE and where changes have been added to the revised manuscript in light of the reviewer comments these are presented in RED.

The author would firstly like to thank the reviewers’ for taking the time to read and critically appraise the manuscript and secondly to thank the reviewers’ for their positive constructive comments in improving the work.

Comments and Suggestions for Authors

Reviewer comments:

The manuscript entitled “Degradation of dental methacrylate-based composites in simulated clinical immersion media” has been reviewed. The results are interesting and helpful. The manuscript needs to be well revised before acceptance. Detailed comments are as follows:

Author’s response:  thank you for your valuable comments and appreciation.

  1. There are some many typo errors in the manuscript. Units should be separated from the numerical value by a space. Please spell-check your manuscript.

Author’s response:  I apologize for all errors and have gone through the entire manuscript accordingly. Please follow marked changes during the manuscript and figures.

  1. The abbreviation of resin-based composites should be RBCs.

Author’s response:  thank you for pointing this out. There were indeed several mistakes with the plural, I apologies for it. Please consider however that in some cased I referred to a singular e.g. RBC fillings – “resin-based composite fillings” and not “resin-based composites fillings”. Similar applies for “RBC restorations” and not “RBCs restorations”.

  1. Some abbreviations, such as FS and E, should not be name twice.

Author’s response:  I carefully replace the full name of the parameters after introducing the abbreviation in the manuscript. Thanks for pointing out this aspect.

  1. Pay attention to unnecessary capitalization of first letters of phrases, such as Urethane dimethacrylate and Triethylene glycol dimethacrylate.

Author’s response:  thank you - the mistake was remediated.

  1. The structuring of 2. Materials and Methods is not adequate and must be improved. The first two sentences are methods (Line 101-104). The paragraph in Line 164-168 should be after the first paragraph of Three-point bending test.

Author’s response:  I placed in the revision the mentioned paragraph lower, as suggested. It just intended to give an overview about what was done, to accommodate the reader with the study design.

  1. Please unify hours and h in the manuscript.

Author’s response:  Thank you for this observation - changes have been made accordingly – only hours was used in the revision.

  1. The footnotes of Table 1 should be removed since these abbreviations are named before and the chemical structures are known.

Author’s response:  in fact, some abbreviations have already been made in the manuscript, as mentioned by the reviewer, but not all.  The Author's Guide always mentions that a table must stand on its own, so all abbreviations in a table must be explained.

  1. In Figure 3, the unit of x-axis is missing.

Author’s response: thank you for the observation – the correction was made accordingly.

  1. In Fig. 4, Hardness should be hardness.

Author’s response: thank you for the observation – the correction was made accordingly.

Round 2

Reviewer 1 Report

Dear authors,

thank you for the renewed possibility to review this interesting manuscript. It has improved significantly and the authors made all necessary changes. 

Two more suggestions:

-Please add the name of the RBCs used in this study to the abstract.

-Table 2 legend is much clearer now. However, I would replace "homogeneous groups" by "groups without significant differences.

Thank you and best wishes!

Author Response

As in the previous revision, all comments to the corresponding author have been addressed independently below. The authors’ rebuttal is always in BLUE and where changes have been added to the revised manuscript in light of the reviewer comments these are presented in RED.

Thanks again for taking the time to read and critically appraise the manuscript.

Comments and Suggestions for Authors

Reviewer comments: thank you for the renewed possibility to review this interesting manuscript. It has improved significantly and the authors made all necessary changes. 

Two more suggestions:

-Please add the name of the RBCs used in this study to the abstract.

 Author’s response:  I followed the above recommendation and added the name of the analysed RBC’s.

-Table 2 legend is much clearer now. However, I would replace "homogeneous groups" by "groups without significant differences.

  Author’s response:  was done accordingly.

Thank you and best wishes!

Author’s response:  thank you!

Reviewer 3 Report

The manuscript has been well revised. It can be accepted if the following comment is considered:

In Table 1,  abbreviations of Bis-GMA, Bis-EMA, TEGDMA, UDMA and DM should be removed since these abbreviations have been named before.

Author Response

As in the previous revision, all comments to the corresponding author have been addressed independently below. The authors’ rebuttal is always in BLUE and where changes have been added to the revised manuscript in light of the reviewer comments these are presented in RED.

Thanks again for taking the time to read and critically appraise the manuscript.

Comments and Suggestions for Authors

The manuscript has been well revised. It can be accepted if the following comment is considered:

In Table 1,  abbreviations of Bis-GMA, Bis-EMA, TEGDMA, UDMA and DM should be removed since these abbreviations have been named before.

Author’s response:  I followed the above recommendation and deleted the abbreviations; Please note that DM has not been abbreviated bevor, so it needed to be mentioned here.